# PHYSICS-AWARE, PROBABILISTIC MODEL ORDER REDUCTION WITH GUARANTEED STABILITY

**Sebastian Kaltenbach, Phaedon-Stelios Koutsourelakis**
Professorship of Continuum Mechanics
Technical University of Munich
{sebastian.kaltenbach,p.s.koutsourelakis}@tum.de

## ABSTRACT

Given (small amounts of) time-series' data from a high-dimensional, fine-grained, multiscale dynamical system, we propose a generative framework for learning an effective, lower-dimensional, coarse-grained dynamical model that is predictive of the fine-grained system's long-term evolution but also of its behavior under different initial conditions. We target fine-grained models as they arise in physical applications (e.g. molecular dynamics, agent-based models), the dynamics of which are strongly non-stationary but their transition to equilibrium is governed by unknown slow processes which are largely inaccessible by brute-force simulations. Approaches based on domain knowledge heavily rely on physical insight in identifying temporally slow features and fail to enforce the long-term stability of the learned dynamics. On the other hand, purely statistical frameworks lack interpretability and rely on large amounts of expensive simulation data (long and multiple trajectories) as they cannot infuse domain knowledge. The generative framework proposed achieves the aforementioned desiderata by employing a flexible prior on the complex plane for the latent, slow processes, and an intermediate layer of physics-motivated latent variables that reduces reliance on data and imbues inductive bias. In contrast to existing schemes, it does not require the a priori definition of projection operators or encoders and addresses simultaneously the tasks of dimensionality reduction and model estimation. We demonstrate its efficacy and accuracy in multiscale physical systems of particle dynamics where probabilistic, long-term predictions of phenomena not contained in the training data are produced.

## 1 INTRODUCTION

High-dimensional, nonlinear systems are ubiquitous in engineering and computational physics. Their nature is in general multi-scale[1]. E.g. in materials, defects and cracks occur on scales of millimeters to centimeters whereas the atomic processes responsible for such defects take place at much finer scales (Belytschko & Song, 2010). Local oscillations due to bonded interactions of atoms (Smit, 1996) take place at time scales of femtoseconds ($10^{-15}s$), whereas protein folding processes which can be relevant for e.g. drug discovery happen at time scales larger than milliseconds ($10^{-3}s$). In Fluid Mechanics, turbulence phenomena are characterized by fine-scale spatiotemporal fluctuations which affect the coarse-scale response (Laizet & Vassilicos, 2009). In all of these cases, macroscopic observables are the result of microscopic phenomena and a better understanding of the interactions between the different scales would be highly beneficial for predicting the system's evolution (Givon et al., 2004). The identification of the different scales, their dynamics and connections however is a non-trivial task and is challenging from the perspective of statistical as well as physical modeling.

---

[1]With the term multiscale we refer to systems whose behavior arises from the synergy of two or more processes occurring at different (spatio)temporal scales. Very often these processes involve different physical descriptions and models (i.e. they are also multi-physics). We refer to the description/model at the finer scale as fine-grained and to the description/model at the coarser scale as coarse-grained.

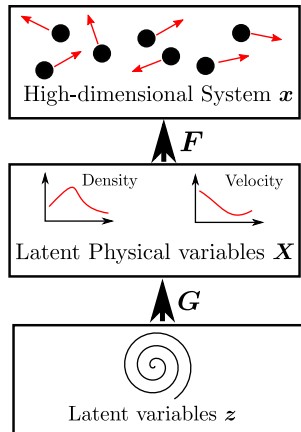

Figure 1: Visual summary of proposed framework. The low-dimensional variables $z$ act via a probabilistic map $G$ as generators of an intermediate layer of latent, physically-motivated variables $X$ that are able to reconstruct the high-dimensional system $x$ with another probabilistic map $F$.

In this paper we propose a novel physics-aware, probabilistic model order reduction framework with guaranteed stability that combines recent advances in statistical learning with a hierarchical architecture that promotes the discovery of interpretable, low-dimensional representations. We employ a generative state-space model with two layers of latent variables. The first describes the latent dynamics using a novel prior on the complex plane that guarantees stability and yields a clear distinction between fast and slow processes, the latter being responsible for the system's long-term evolution. The second layer involves physically-motivated latent variables which infuse inductive bias, enable connections with the very high-dimensional observables and reduce the data requirements for training. The probabilistic formulation adopted enables the quantification of a crucial, and often neglected, component in any model compression process, i.e. the predictive uncertainty due to information loss. We finally want to emphasize that the problems of interest are *Small Data* ones due to the computational expense of the physical simulators. Hence the number of time-steps as well as the number of time-series used for training is small as compared to the dimension of the system and to the time-horizon over which predictions are sought.

## 2 Physics-Aware, Probabilistic Model Order Reduction

Our data consists of $N$ times-series $\{x_{0:T}^{(i)}\}_{i=1}^{N}$ over $T$ time-steps generated by a computational physics simulator. This can represent positions and velocities of each particle in a fluid or those of atoms in molecular dynamics. Their dimension is generally very high i.e. $x_t \in \mathcal{M} \subset \mathbb{R}^f$ ($f >> 1$). In the context of state-space models, the goal is to find a lower-dimensional set of collective variables or latent generators $z_t$ and their associated dynamics. Given the difficulties associated with these tasks and the solutions that have been proposed in statistics and computational physics literature, we advocate the use of an intermediate layer of physically-motivated, lower-dimensional variables $X_t$ (e.g. density or velocity fields), the meaning of which will become precise in the next sections. These variables provide a coarse-grained description of the high-dimensional observables and imbue interpretability in the learned dynamics. Using $X_t$ alone (without $z_t$) would make it extremely difficult to enforce long-term stability (see Appendix H.2) while ensuring sufficient complexity in the learned dynamics (Felsberger & Koutsourelakis, 2019; Champion et al., 2019). Furthermore and even if the dynamics of $x_t$ are first-order Markovian, this is not necessarily the case for $X_t$ (Chorin & Stinis, 2007). The latent variables $z_t$ therefore effectively correspond to a nonlinear coordinate transformation that yields not only Markovian but also stable dynamics (Gin et al., 2019). The general framework is summarized in Figure 1 and we provide details in the next section.

## 2.1 MODEL STRUCTURE

Our model consists of three levels. At the first level, we have the latent variables $z_t$ which are connected with $X_t$ in the second layer through a probabilistic map $G$. The physical variables $X_t$ are finally connected to the high-dimensional observables through another probabilistic map $F$. We parametrize $F$, $G$ with deep neural networks and denote by $\theta_1$ and $\theta_2$ the corresponding parameters (see Appendix D). In particular, we postulate the following relations:

$$z_{t,j} = z_{t-1,j} \exp(\lambda_j) + \sigma_j \epsilon_{t,j} \quad \lambda_j \in \mathbb{C}, \quad \epsilon_{t,j} \sim \mathcal{CN}(0,1), \; j = 1, 2, \ldots, h \quad (1)$$

$$X_t = G(z_t, \theta_1) \quad (2)$$

$$x_t = F(X_t, \theta_2) \quad (3)$$

We assume that the latent variables $z_t$ are complex-valued and a priori independent. Complex variables were chosen as their evolution includes a harmonic components which are observed in many physical systems. In Appendix H.1 we present results with a real-valued latent variables $z_{t,j}$ and illustrate their limitations. We model their dynamics with a discretized Ornstein-Uhlenbeck process on the complex plane with initial conditions $z_{0,j} \sim \mathcal{CN}(0, \sigma_{0,j}^2)^2$. The parameters associated with this level are denoted summarily by $\theta_0 = \{\sigma_{0,j}^2, \sigma_j^2, \lambda_j\}_{j=1}^h$. These, along with $\theta_1, \theta_2$ mentioned earlier, and the state variables $X_t$ and $z_t$ have to be inferred from the data $x_t$. We explain each of the aforementioned components in the sequel.

### 2.1.1 STABLE LOW-DIMENSIONAL DYNAMICS

While the physical systems (e.g. particle dynamics) of interest are highly non-stationary, they generally converge to equilibrium in the long-term. We enforce long-term stability here by ensuring that the real-part of the $\lambda_j$'s in Equation (1) is negative, i.e.:

$$\lambda_j = \Re(\lambda_j) + i \, \Im(\lambda_j) \text{ with } \Re(\lambda_j) < 0 \quad (4)$$

which guarantees first and second-order stability i.e. the mean as well as the variance are bounded at all time steps.

The transition density each process $z_{t,j}$ is given by:

$$p\left(z_{t,j} \mid z_{t-1,j}\right) = \mathcal{N}\left( \begin{bmatrix} \Re(z_{t,j}) \\ \Im(z_{t,j}) \end{bmatrix} \; \middle| \; s_j \, R_j \begin{bmatrix} \Re(z_{t-1,j}) \\ \Im(z_{t-1,j}) \end{bmatrix}, \; I \frac{\sigma_j^2}{2} \right) \quad (5)$$

where the orthogonal matrix $R_j$ depends on the imaginary part of $\lambda_j$:

$$R_j = \begin{bmatrix} \cos(\Im(\lambda_j)) & -\sin(\Im(\lambda_j)) \\ \sin(\Im(\lambda_j)) & \cos(\Im(\lambda_j)) \end{bmatrix} \quad (6)$$

and the decay rate $s_j$ depends on the real part of $\lambda_j$:

$$s_j = \exp(\Re(\lambda_j)) \quad (7)$$

i.e. the closer to zero the latter is, the "slower" the evolution of the corresponding process is. As in probabilistic Slow Feature Analysis (SFA) (Turner & Sahani, 2007; Zafeiriou et al., 2015), we set $\sigma_j^2 = 1 - \exp(2 \, \Re(\lambda_j)) = 1 - s_j^2$ and $\sigma_{0,j}^2 = 1$. As a consequence, a priori, the latent dynamics are stationary[3] and an ordering of the processes $z_{t,j}$ is possible on the basis of $\Re(\lambda_j)$. Hence the only independent parameters are the $\lambda_j$, the imaginary part of which can account for periodic effects in the latent dynamics (see Appendix B).

The joint density of $z_t$ can finally be expressed as:

$$p(z_{0:T}) = \prod_{j=1}^h \left( \prod_{t=1}^T p(z_{t,j} \mid z_{t-1,j}, \theta_0) p(z_{0,j} | \theta_0) \right) \quad (8)$$

The transition density between states at non-neighbouring time-instants is also available analytically and is useful for training on longer trajectories or in cases of missing data. Details can be found in Appendix B.

---

[2]A short review of complex normal distributions, denoted by $\mathcal{CN}$, can be found in Appendix A.

[3]More details can be found in Appendix B.

### 2.1.2 PROBABILISTIC GENERATIVE MAPPING

We employ fully probabilistic maps between the different layers which involve two conditional densities based on Equations (2) and (3), i.e.:

$$p(\boldsymbol{x}_t \mid \boldsymbol{X}_t, \boldsymbol{\theta}_2) \quad \text{and} \quad p(\boldsymbol{X}_t \mid \boldsymbol{z}_t, \boldsymbol{\theta}_1) \qquad (9)$$

In contrast to the majority of physics-motivated papers (Chorin & Stinis, 2007; Champion et al., 2019) as well as those based on transfer-operators Klus et al. (2018), we note that the generative structure adopted does not require the prescription of a restriction operator (or encoder) and the reduced variables need not be selected a priori but rather are adapted to best reconstruct the observables.

The splitting of the generative mapping into two parts through the introduction of the intermediate variables $\boldsymbol{X}_t$ has several advantages. Firstly, known physical dependencies between the data $\boldsymbol{x}$ and the physical variables $\boldsymbol{X}$ can be taken into account, which reduces the complexity of the associated maps and the total number of parameters. For instance, in the case of particle simulations where $\boldsymbol{X}$ represents a density or velocity field, i.e. it provides a coarsened or averaged description of the fine-scale observables, it can be used to (probabilistically) reconstruct the positions or velocities of the particles. This physical information can be used to compensate for the lack of data when only few training sequences are available (Small data) and can seen as a strong prior to the model order reduction framework. Due to the lower dimension of associated variables, the generative map between $\boldsymbol{z}_t$ and $\boldsymbol{X}_t$ can be more easily learned even with few training samples. Lastly, the inferred physical variables $\boldsymbol{X}$ can provide insight and interpretability to the analysis of the physical system.

### 2.2 INFERENCE AND LEARNING

Given the probabilistic relations above, our goal is to infer the state variables $\boldsymbol{X}_{0:T}^{(1:n)}, \boldsymbol{z}_{0:T}^{(1:n)}$ as well as all model parameters $\boldsymbol{\theta}$. We follow a hybrid Bayesian approach in which the posterior of the state variables is approximated using structured Stochastic Variational Inference (Hoffman et al., 2013) and MAP point estimates for $\boldsymbol{\theta} = \{\boldsymbol{\theta}_0, \boldsymbol{\theta}_1, \boldsymbol{\theta}_2\}$ are computed.

The application of Bayes' rule leads to the following posterior:

$$p(\boldsymbol{X}_{0:T}^{(1:n)}, \boldsymbol{z}_{0:T}^{(1:n)}, \boldsymbol{\theta} | \boldsymbol{x}_{0:T}^{(1:n)}) = \frac{p(\boldsymbol{x}_{0:T}^{(1:n)} | \boldsymbol{X}_{0:T}^{(1:n)}, \boldsymbol{z}_{0:T}^{(1:n)}, \boldsymbol{\theta}) \, p(\boldsymbol{X}_{0:T}^{(1:n)}, \boldsymbol{z}_{0:T}^{(1:n)}, \boldsymbol{\theta})}{p(\boldsymbol{x}_{0:T}^{(1:n)})} \qquad (10)$$

$$= \frac{p(\boldsymbol{x}_{0:T}^{(1:n)} | \boldsymbol{X}_{0:T}^{(1:n)}, \boldsymbol{\theta}) \, p(\boldsymbol{X}_{0:T}^{(1:n)} \mid \boldsymbol{z}_{0:T}^{(1:n)}, \boldsymbol{\theta}) \, p(\boldsymbol{z}_{0:T}^{(1:n)} \mid \boldsymbol{\theta}) \, p(\boldsymbol{\theta})}{p(\boldsymbol{x}_{0:T}^{(1:n)})} \qquad (11)$$

where $p(\boldsymbol{\theta})$ denotes the prior on the model parameters. In the context of variational inference, we use the following factorization of the approximate posterior[4]:

$$q_{\boldsymbol{\phi}}(\boldsymbol{X}_{0:T}^{(1:n)}, \boldsymbol{z}_{0:T}^{(1:n)}) = \prod_{i=1}^{n} \left( \prod_{j=0}^{h} q_{\boldsymbol{\phi}}(\boldsymbol{z}_{0:T,j}^{(i)} \mid \boldsymbol{X}_{0:T}^{(i)}) \right) q_{\boldsymbol{\phi}}(\boldsymbol{X}_{0:T}^{(i)}) \qquad (12)$$

We approximate the conditional posterior of $\boldsymbol{z}$ given $\boldsymbol{X}$ with a complex multivariate normal which is parameterized using a tridiagonal precision matrix as proposed in Archer et al. (2015); Bamler & Mandt (2017). This retains dependencies between temporally neighbouring $\boldsymbol{z}$, but the number of parameters grows linearly with the dimension of $\boldsymbol{z}$ which leads to a highly scalable algorithm. For the variational posterior of $\boldsymbol{X}$ we employ a Gaussian with a diagonal covariance, i.e.:

$$q_{\boldsymbol{\phi}}(\boldsymbol{z}_{0:T,j}^{(i)} \mid \boldsymbol{X}_{0:T}^{(i)}) = \mathcal{CN}(\boldsymbol{\mu}_{\boldsymbol{\phi}}(\boldsymbol{X}_{0:T}^{(i)}), \left[ \boldsymbol{B}_{\boldsymbol{\phi}}(\boldsymbol{X}_{0:T}^{(i)}) \boldsymbol{B}_{\boldsymbol{\phi}}(\boldsymbol{X}_{0:T}^{(i)})^T \right]^{-1}) \qquad q_{\boldsymbol{\phi}}(\boldsymbol{X}_{0:T}^{(i)}) = \mathcal{N}(\boldsymbol{\mu}_{\boldsymbol{\phi}}^{(i)}, \boldsymbol{\Sigma}_{\boldsymbol{\phi}}^{(i)})$$
$$(13)$$

We denote summarily with $\boldsymbol{\phi}$ the parameters involved and note that deep neural networks are used for the mean $\boldsymbol{\mu}_{\boldsymbol{\phi}}(\boldsymbol{X}_{0:T}^{(i)})$ as well as the upper bidiagonal matrix $\boldsymbol{B}_{\boldsymbol{\phi}}(\boldsymbol{X}_{0:T}^{(i)})$. Details on the neural net architectures employed are provided in Section 4 and in Appendix D.

---

[4]We note that this factorization does not introduce any error due to the conditional independence of $\boldsymbol{x}, \boldsymbol{z}$ given $\boldsymbol{X}$.

It can be readily shown that the optimal parameter values are found by maximizing the Evidence Lower Bound (ELBO) $\mathcal{F}(q_\phi(\boldsymbol{X}_{0:T}^{(1:n)}, \boldsymbol{z}_{0:T}^{(1:n)}), \boldsymbol{\theta})$ which is derived in Appendix C. We compute Monte Carlo estimates of the gradient of the ELBO with respect to $\phi$ and $\boldsymbol{\theta}$ with the help of the reparametrization trick (Kingma & Welling, 2013) and carry out stochastic optimization with the ADAM algorithm (Kingma & Ba, 2014).

## 2.3 PREDICTIONS

Once state variables have been inferred and MAP estimates $\boldsymbol{\theta}^{MAP}$ for the model parameters have been obtained, the reduced model can be used for probabilistic future predictions. In order to do so for a time sequence used in training, we employ the following Monte Carlo scheme to generate a sample $\boldsymbol{x}_{T+P}$, i.e. $P$ time-steps into the future:

1. Sample $\boldsymbol{X}_T$ and $\boldsymbol{z}_T$ from the inferred posterior $q_\phi(\boldsymbol{z}_{0:T} \mid \boldsymbol{X}_{0:T})q_\phi(\boldsymbol{X}_{0:T})$.
2. Propagate $\boldsymbol{z}_T$ for $P$ time steps forward by using the conditional density in Equation (5).
3. Sample $\boldsymbol{X}_{T+P}$ and $\boldsymbol{x}_{T+P}$ from $p(\boldsymbol{X}_{T+P} \mid \boldsymbol{z}_{T+P}, \boldsymbol{\theta}_1^{MAP})$ and $p(\boldsymbol{x}_{T+P} \mid \boldsymbol{X}_{T+P}, \boldsymbol{\theta}_2^{MAP})$ respectively.

More importantly perhaps, the trained model can be used for predictions under new initial conditions, e.g. $\boldsymbol{x}_0$. To achieve this, first the posterior $p(\boldsymbol{z}_0|\boldsymbol{x}_0) \propto \int p(\boldsymbol{x}_0|\boldsymbol{X}_0, \boldsymbol{\theta}_2^{MAP}) \, p(\boldsymbol{X}_0 \mid \boldsymbol{z}_0, \boldsymbol{\theta}_1^{MAP}) \, d\boldsymbol{X}_0$ must be found before the Monte Carlo steps above can be employed starting at $T = 0$.

## 3 RELATED WORK

The main theme of our work is the learning of low-dimensional dynamical representations that are stable, interpretable and make use of physical knowledge.

**Linear latent dynamics:** In this context, the line of work that most closely resembles ours pertains to the use of Koopman-operator theory (Koopman, 1931) which attempts to identify appropriate transformations of the original coordinates that yield linear dynamics (Klus et al., 2018). We note that these approaches (Lusch et al., 2018; Champion et al., 2019; Gin et al., 2019; Lee & Carlberg, 2020) require additionally the specification of an encoder i.e. a map from the original description to the reduced coordinates which we avoid in the generative formulation adopted. Furthermore only a small fraction are probabilistic and can quantify predictive uncertainties but very often employ restrictive parametrizations for the Koopman matrix in order to ensure long-term stability (Pan & Duraisamy, 2020). To the best of our knowledge, none of the works along these lines employ additional, physically-motivated variables and as a result have demonstrated their applicability only in lower-dimensional problems and require very large amounts of training data or some ad hoc preprocessing. We provide comparative results with Koopman-based deterministic and probabilistic models in Appendix H.3.

**Data-driven discovery of nonlinear dynamics:** The data-driven discovery of governing dynamics has received tremendous attention in recent years. Efforts based on the Mori-Zwanzig formalism can accurately identify dynamics of pre-defined variables, which also account for memory effects, but cannot reconstruct the full fine-grained picture or make predictions about other quantities of interest (Chorin & Stinis, 2007; Kondrashov et al., 2015; Ma et al., 2019). Similar restrictions apply when neural-network-based models are employed as e.g. (Chen et al., 2018; Li et al., 2020). Efforts based on the popular SINDy algorithm (Brunton et al., 2016) require additionally data of the time-derivatives of the variables of interest which when estimated with finite-differences introduce errors and reduce robustness. Sparse Bayesian learning tools in combination with physically-motivated variables and generative models have been employed by (Felsberger & Koutsourelakis, 2019) but cannot guarantee the long-term stability of the learned dynamics as we also show in Appendix H.2 and in the context of the systems investigated in section 4.

**Infusing domain knowledge from physics:** Several efforts have been directed in endowing neural networks with invariances or equivariances arising from physical principles. Usually those pertain to translation or rotation invariance and are domain-specific as in Schütt et al. (2017). More general formulations such as Hamiltonian (Greydanus et al., 2019; Toth et al., 2019) and Lagrangian

Dynamics (Lutter et al., 2019) are currently restricted in terms of the dimension of the dynamical system. Physical knowledge has been exploited in conjunction with Gaussian Processes in (Camps-Valls et al., 2018) as well as in the context of PDEs for constructing reduced-order models as in (Grigo & Koutsourelakis, 2019) or for learning modulated derivatives using Graph Neural Networks as in (Seo et al., 2020). Another approach involves using physical laws as regularization terms or for augmenting the loss function as in (Raissi et al., 2019; Lusch et al., 2018; Zhu et al., 2019; Kaltenbach & Koutsourelakis, 2020). In the context of molecular dynamics multiple schemes for coarse-graining which also guarantee long-term stability have been proposed by Noé (2018) and Wu et al. (2017; 2018). In our formulation, physically-motivated latent variables are used to facilitate generative maps to very high-dimensional data and serve as the requisite information bottleneck in order to reduce the amount of training data needed.

**Slowness and interpretability:** Finally, in contrast to general state-space models for analyzing time-series data such as Karl et al. (2016); Rangapuram et al. (2018); Li et al. (2019), the prior proposed on the complex-valued $z_t$ enable the discovery of slow features which are crucial in predicting the evolution of multiscale systems and in combination with the variables $X_t$ can provide interpretability and insight into the underlying physical processes.

## 4 EXPERIMENTS

The high-dimensional, fine-grained model considered consists of $f$ identical particles which can move in the bounded one-dimensional domain $s \in [-1, 1]$ (under periodic boundary conditions). The variables $x_t$ consist therefore of the coordinates of the particles at each time instant $t$ and the dimension of the system $f$ is equal to the number of particles. We consider two types of stochastic particle dynamics that correspond to an advection-diffusion-type (section 4.1) and a viscous-Burgers'-type (section 4.2) behavior. In all experiments, the physically-motivated variables $X_t$ relate to a discretization of the particle density into $d = 25$ equally-sized bins for advection-diffusion-type dynamics and into $d = 64$ equally-sized bins for viscous-Burgers'-type dynamics. In order to automatically enforce the conservation of mass at each time instant, we make use of the softmax function i.e. the particle density at a bin $k$ is expressed as $\frac{\exp(X_{t,k})}{\sum_{l=0}^{d} \exp(X_{t,l})}$. Given this, the probabilistic map ($F$ in Equation (3)) corresponds to a multinomial density, i.e.:

$$p(\boldsymbol{x}_t | \boldsymbol{X}_t) = \frac{f!}{m_1(\boldsymbol{x}_t)! \, m_2(\boldsymbol{x}_t)! \ldots m_k(\boldsymbol{x}_t)!} \prod_{k=1}^{d} \left( \frac{\exp(X_{t,k})}{\sum_{l=0}^{d} \exp(X_{t,l})} \right)^{m_k(\boldsymbol{x}_t)} \tag{14}$$

where $m_k(\boldsymbol{x}_t)$ is the number of particles in bin $k$. The underlying assumption is that, given $\boldsymbol{X}_t$, the coordinates of the particles $\boldsymbol{x}_t$ are *conditionally* independent. This does *not* imply that they move independently nor that they cannot exhibit coherent behavior (Felsberger & Koutsourelakis, 2019). Furthermore, the aforementioned model automatically satisfies permutation-invariance which a central feature of the fine-grained dynamics. This is another advantage of the physically motivated intermediate variables $\boldsymbol{X}$ as enforcing such symmetries/invariances is not a trivial task even when highly expressive models (e.g. neural networks) are used (Rezende et al., 2019). The practical consequence of Equation (14) is that no parameters $\boldsymbol{\theta}_2$ need to be inferred.

The second map pertains to $p(\boldsymbol{X}_t \mid \boldsymbol{z}_t, \boldsymbol{\theta}_1)$ which we represent with a multivariate normal distribution with a mean and a diagonal covariance matrix modeled by a neural network with parameters $\boldsymbol{\theta}_1$. Details of the parameterization can be found in Appendix D.

We assess the performance of the method by computing first- and second-order statistics as illustrated in the sequel as well as in Appendix F. We provide comparative results on the same examples in Appendix H where we report on the performance of various alternatives, such as a formulation without the latent variables $\boldsymbol{z}_t$ as well as deterministic and probabilistic Koopman-based models. Moreover a short study on the effect of the amount of training data is included in Appendix G.

### 4.1 PARTICLE DYNAMICS: ADVECTION-DIFFUSION

We train the model on $N = 64$ time-series of the positions of $f = 250 \times 10^3$ particles over $T = 40$ time-steps which were simulated as described in Appendix D.1. Furthermore, we made

use of $h = 5$ complex, latent processes $z_{t,j}$. Details regarding the variational posteriors and neural network architectures involved can be found in the Appendix D.1.

In Figure 2, the estimates of the complex-valued parameters $\lambda_j$ are plotted as well as the inferred and predicted time-evolution of 2 associated processes $z_{t,j}$ on the complex plane. We note the clear separation of time-scales in the first plot with two slow processes, one intermediate and two fast ones. This is also evident in the indicative trajectories on the complex plane. A detailed discussion of the map $G$ learned (Equation (2)) between $z_t$ and $X_t$ can be found in the Appendix E.

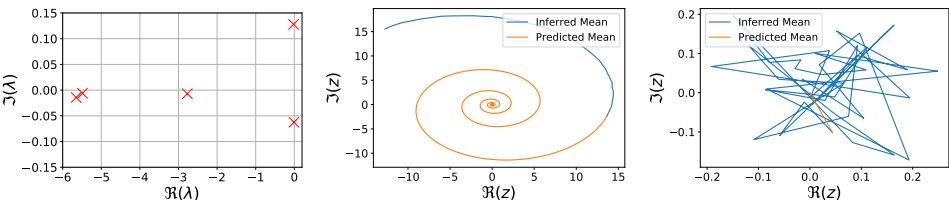

Figure 2: Estimated $\lambda_j$ (left) and the time evolution of two $z_{t,j}$ processes where one is slow (middle) and the other fast (right).

In Figure 3 we compare the true particle density with the one predicted by the trained reduced model. We note that the latter is computed by reconstructing the $x_t$ futures. We observe that the model is able to accurately track first-order statistics well into the future.

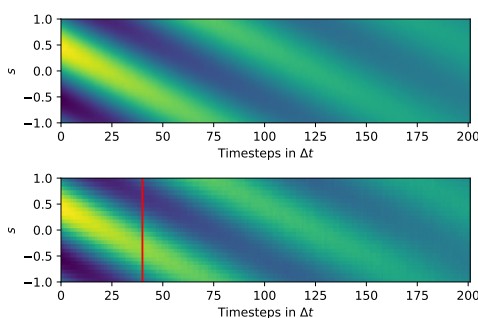

Figure 3: Particle density: Inferred and predicted posterior mean (bottom) in comparison with the ground truth (top). The red line divides inferred quantities from predicted ones. The horizontal axis corresponds to time-steps and the vertical to the one-dimensional spatial domain of the problem $s \in [-1, 1]$.

A more detailed view of the predictive estimates with snapshots of the particle density at selected time instances is presented in Figure 4. Here, not only the posterior mean but also the associated uncertainty is displayed. We want to emphasize the last Figure at $t = 1000$ when the steady state has been reached which clearly shows that our model is capable of converging to stable equilibrium.

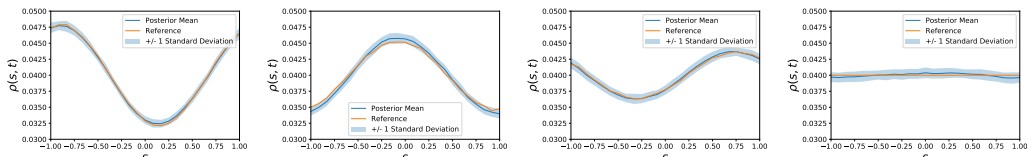

Figure 4: Predicted particle density profiles at $t = 80, 120, 160, 1000$ (from left to right).

Since the proposed model is capable of probabilistically reconstructing the whole fine-grained picture, i.e. $x_t$, predictions with regards to any observable can be obtained. In Appendix F.1 we assess the accuracy of predictions in terms of second-order statistics, and in particular for the probability of finding simultaneously a pair of particles at two specified positions.

Finally, we demonstrate the accuracy of the trained in model in producing predictions under *unseen* initial conditions as described in section 2.3. Figure 5 depicts a new initial condition in terms of the particle density according to which particle positions $x_0$ were drawn. The posterior on the corresponding $z_0$ (see section 2.3) can be used to reconstruct the density as shown also on the same Figure.

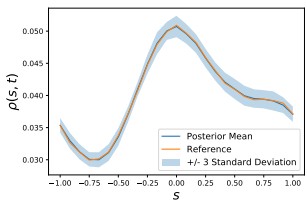

Figure 5: New initial condition (not used in training) in terms of the particle density. Reference shows the actual initial condition, whereas the posterior mean and uncertainty bounds correspond to the reconstruction of the initial condition based on the inferred latent variables $z_0$.

Figure 6 shows predictions of the particle density (i.e. first-order statistics) at various timesteps. We want to emphasise the frame on the right, for $t = 500$ which shows the steady state of the system. We note that even though the initial condition was not contained in the training data, our framework is able to correctly track the system's evolution and to predict the correct steady state.

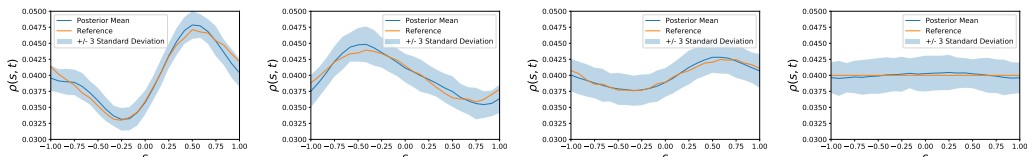

Figure 6: Predictions of the particle density at $t = 25, 75, 125, 500$ (left to right) for on the new initial condition in Figure 5.

## 4.2 PARTICLE DYNAMICS: VISCOUS BURGERS' EQUATION

In this example, we made use of $N = 64$ sequences of $f = 500 \times 10^3$ particles over $T = 40$ time-steps. Details regarding the physical simulator, the stochastic interactions between particles as well as the associated network architectures are contained in Appendix D.2. As in the previous example, we employed the particle density with the softmax transformation for $X_t$ and $h = 5$ complex-valued processes $z_t$ at the lowest model level.

In Figure 7 the estimated values for $\lambda_j$ are shown where the clear separation of time-scales with three slow and two fast processes can be observed.

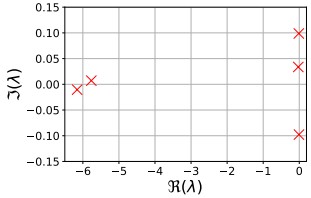

Figure 7: Estimated $\lambda_j$ for the viscous Burgers' system.

In Figure 8 we compare the evolution of the true particle density with the (posterior mean of) the model-predicted one. We point out the sharp front at the lower left corner which is characteristic of the Burgers' equation and which eventually dissipates due to the viscosity. This is captured in the inferred as well as in the predicted solution.

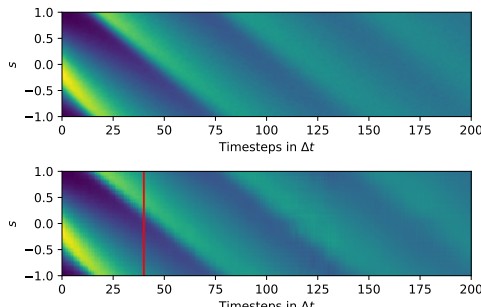

Figure 8: Particle density: Inferred and predicted posterior mean (bottom) in comparison with the ground truth (top). The red line divides inferred quantities from predicted ones. The horizontal axis corresponds to time-steps and the vertical to the one-dimensional spatial domain of the problem $s \in [-1, 1]$

A more detailed view on the predictive results with snapshots of the particle density at selected time instances is presented in Figure 9. We emphasize again the stable convergence of the learned dynamics to the steady state as well as the accuracy in capturing, propagating and dissipating the shock front.

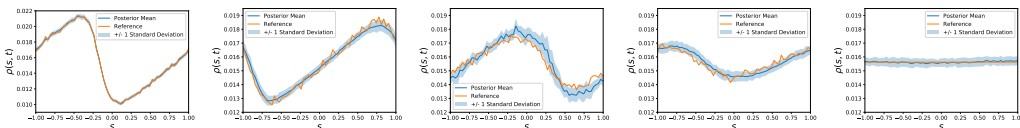

Figure 9: Predicted particle density profiles at $t = 40, 80, 120, 160, 1000$ (from left to right).

We compare the accuracy of the predictions for second-order statistics of the fine-grained system in terms of the two-particle probability in Appendix F.2 where excellent agreement with the ground truth, i.e. the one computed by simulating the fine-grained system, is observed.

## 5    CONCLUSIONS

We presented a framework for efficiently learning a lower-dimensional, dynamical representation of a high-dimensional, fine-grained system that is predictive of its long-term evolution and whose stability is guaranteed. We infuse domain knowledge with the help of an additional layer of latent variables. The latent variables at the lowest level provide an interpretabable separation of the time-scales and ensure the long-term stability of the learned dynamics. We employed scalable variational inference techniques and applied the proposed model in data generated from very large systems of interacting particles. In all cases accurate probabilistic predictions were obtained both in terms of first- and second-order statistics over a very long time range into the future. More importantly, the ability of the trained model to produce predictions under new, unseen initial conditions was demonstrated. An obvious limitation for the applicability of the proposed method to general dynamical systems pertains to the physically motivated variables $\boldsymbol{X}$. While such variables are available for several classes of physical systems, they need to be re-defined when moving to new problems and expert elicitation might be necessary. Furthermore, if an incomplete list of such variables is available from physical insight, this would need to be complemented by additional variables discovered from data. To that end, one could envision that some of the abstract latent variables $\boldsymbol{z}_t$ or functions thereof, e.g. $\boldsymbol{f}(\boldsymbol{z}_t)$, could be employed in a generative map of the form $\boldsymbol{x}_t = \boldsymbol{F}(\boldsymbol{X}_t, \boldsymbol{f}(\boldsymbol{z}_t))$ instead of Equation (3). A final deficiency of the proposed model is the lack of an automated procedure for determining the appropriate number of $\boldsymbol{z}$. We believe that the ELBO, which provides a lower bound on the model evidence, could be used for this purpose.

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

## A  COMPLEX NORMAL DISTRIBUTION

In this Appendix, the complex random normal distribution is reviewed. The mathematical definitions introduced follow Andersen et al. (1995):

A $p$-variate complex normal random variable $\boldsymbol{Y} \in \mathbb{C}^p$ with $\boldsymbol{Y} \sim \mathcal{CN}(\boldsymbol{\mu}_{\mathbb{C}}, \boldsymbol{\Sigma}_{\mathbb{C}})$ is defined by a complex mean vector $\boldsymbol{\mu}_{\mathbb{C}} \in \mathbb{C}^p$ and a complex Covariance Matrix $\boldsymbol{\Sigma}_{\mathbb{C}} \in \mathbb{C}^{p \times p}_+$. The density with respect to Lesbegue measures on $\mathbb{C}^p$ can be stated as:

$$f_{\boldsymbol{Y}}(\boldsymbol{y}) = \pi^{-p} \det(\boldsymbol{\Sigma}_{\mathbb{C}})^{-1} \exp\left(-(\boldsymbol{y} - \boldsymbol{\mu}_{\mathbb{C}})^* \boldsymbol{\Sigma}_{\mathbb{C}}^{-1} (\boldsymbol{y} - \boldsymbol{\mu}_{\mathbb{C}})\right) \tag{15}$$

where $*$ indicates the conjugate transpose of a matrix.

This complex normal random variable has similar properties to the well-known, real-valued counterpart. For instance, linear transformations of complex random normal variables are again complex random normal variables.

These properties directly follow from the fact, that for a complex random normal variable there exists an isomorphic transformation to a real valued $2p$-variate normal random variable $W \in \mathbb{R}^{2p}$. This random normal variable is defined with mean

$$\boldsymbol{\mu}_{\mathbb{R}} = \begin{bmatrix} \Re(\boldsymbol{\mu}_{\mathbb{C}}) \\ \Im(\boldsymbol{\mu}_{\mathbb{C}}) \end{bmatrix} \tag{16}$$

and covariance

$$\boldsymbol{\Sigma}_{\mathbb{R}} = \frac{1}{2} \begin{bmatrix} \Re(\boldsymbol{\Sigma}_{\mathbb{C}}) & -\Im(\boldsymbol{\Sigma}_{\mathbb{C}}) \\ \Im(\boldsymbol{\Sigma}_{\mathbb{C}}) & \Re(\boldsymbol{\Sigma}_{\mathbb{C}}) \end{bmatrix} \tag{17}$$

Therefore: $W \sim \mathcal{N}(\boldsymbol{\mu}_{\mathbb{R}}, \boldsymbol{\Sigma}_{\mathbb{R}})$

As an example the real valued isomorphic counterpart of the standard complex random normal distribution $\mathcal{CN}(0, 1)$ is the bivariate normal distribution $\mathcal{N}(\boldsymbol{0}, \frac{1}{2}\mathbf{I})$.

## B  CHOICE OF VARIANCE FOR A-PRIORI STEADY STATE

We derive transient and stationary properties of the complex-valued latent processes $z_{t,j}$ and justify our choices for the model parameters. Based on Equation (1) and for each process $j$, the dynamics can be written in terms of the real and imaginary parts in two dimensions as follows:

$$
\begin{bmatrix} \Re(z_{t,j}) \\ \Im(z_{t,j}) \end{bmatrix} = \boldsymbol{A}_j \begin{bmatrix} \Re(z_{t,j-1}) \\ \Im(z_{t,j-1}) \end{bmatrix} + \sigma_j \begin{bmatrix} \Re(\epsilon_{t,j}) \\ \Im(\epsilon_{t,j}) \end{bmatrix}, \tag{18}
$$

or:

$$
\begin{bmatrix} \Re(z_{t,j}) \\ \Im(z_{t,j}) \end{bmatrix} = \boldsymbol{A}_j^t \begin{bmatrix} \Re(z_{t,0}) \\ \Im(z_{t,0}) \end{bmatrix} + \sigma_j \sum_{k=0}^{t-1} \boldsymbol{A}_j^k \begin{bmatrix} \Re(\epsilon_{t-k,j}) \\ \Im(\epsilon_{t-k,j}) \end{bmatrix} \tag{19}
$$

where $\Re(\epsilon_{t,j}), \Im(\epsilon_{t,j}) \sim \mathcal{N}(0, 1/2)$. The matrix $\boldsymbol{A}_j$ is given by (see also Equations (6), (7)):

$$
\boldsymbol{A}_j = s_j \boldsymbol{R}_j = e^{\Re(\lambda_j)} \begin{bmatrix} \cos(\Im(\lambda_j)) & -\sin(\Im(\lambda_j)) \\ \sin(\Im(\lambda_j)) & \cos(\Im(\lambda_j)) \end{bmatrix} \tag{20}
$$

and can be diagonalized as $\boldsymbol{A}_j = \boldsymbol{V} \boldsymbol{P}_j \boldsymbol{V}^*$ where:

$$
\boldsymbol{V} = \frac{1}{\sqrt{2}} \begin{bmatrix} 1 & 1 \\ -i & i \end{bmatrix}, \qquad \boldsymbol{P}_j = \begin{bmatrix} e^{\lambda_j} & 0 \\ 0 & e^{\bar{\lambda}_j} \end{bmatrix}, \tag{21}
$$

$\boldsymbol{V}^*$ is the conjugate transpose of $\boldsymbol{V}$ and $\bar{\lambda}_j$ denotes the complex conjugate of $\lambda_j$. Due to the linearity of the model and the Gaussian initial conditions, i.e. $\Re(z_{0,j}), \Im(z_{0,j}) \sim \mathcal{N}(0, \sigma_{0,j}^2/2)$, the marginal will remain Gaussian at all times $t$. A direct consequence of the above is that the mean of real and imaginary parts is always zero, i.e.:

$$
\boldsymbol{\mu}_t^{(j)} = \begin{bmatrix} \mathbb{E}[\Re(z_{t,j})] \\ \mathbb{E}[\Im(z_{t,j})] \end{bmatrix} = \begin{bmatrix} 0 \\ 0 \end{bmatrix}. \tag{22}
$$

Furthermore, the covariance $\boldsymbol{C}_t^{(j)}$ is given by:

$$
\boldsymbol{C}_t^{(j)} = \mathbb{E}\left[ \begin{bmatrix} \Re(z_{t,j}) \\ \Im(z_{t,j}) \end{bmatrix} [\Re(z_{t,j})\, \Im(z_{t,j})] \right] = \frac{\sigma_{0,j}^2}{2} \boldsymbol{A}_j^t (\boldsymbol{A}_j^T)^t + \sigma_k^2 \sum_{k=0}^{t-1} \boldsymbol{A}_j^k (\boldsymbol{A}_j^T)^k \tag{23}
$$

which upon use of the diagonalized form of $\boldsymbol{A}_j$ yields:

$$
\boldsymbol{C}_t^{(j)} = \frac{\sigma_{0,j}^2}{2} \boldsymbol{V} \begin{bmatrix} 0 & e^{2t\Re(\lambda_j)} \\ e^{2t\Re(\lambda_j)} & 0 \end{bmatrix} \boldsymbol{V}^T + \frac{\sigma_j^2}{2} \boldsymbol{V} \begin{bmatrix} 0 & \frac{1-e^{2t\Re(\lambda_j)}}{1-e^{2\Re(\lambda_j)}} \\ \frac{1-e^{2t\Re(\lambda_j)}}{1-e^{2\Re(\lambda_j)}} & 0 \end{bmatrix} \boldsymbol{V}^T \tag{24}
$$

As mentioned earlier, a necessary condition for the long-term stability of the processes is that $\Re(\lambda_j) < 0$, in which case and as $t \to \infty$ it leads to:

$$
\boldsymbol{C}_t^{(j)} \to \boldsymbol{C}_\infty^{(j)} = \frac{\sigma_j^2}{2} \boldsymbol{V} \begin{bmatrix} 0 & \frac{1}{1-e^{2\Re(\lambda_j)}} \\ \frac{1}{1-e^{2\Re(\lambda_j)}} & 0 \end{bmatrix} \boldsymbol{V}^T = \frac{\sigma_j^2}{2(1-e^{2\Re(\lambda_j)})} \boldsymbol{I} \tag{25}
$$

This implies that real and imaginary parts are asymptotically uncorrelated with a variance $\frac{\sigma_j^2}{2(1-e^{2\Re(\lambda_j)})}$. By setting $\sigma_j^2 = 1 - e^{2\Re(\lambda_j)}$ we enable direct comparisons between $z_{t,j}$ solely on the basis of $Re(\lambda_j)$ i.e. the degree of slowness (Turner & Sahani, 2007; Zafeiriou et al., 2015)). In this case the asymptotic variance of real and imaginary parts becomes $1/2$. Finally by setting $\sigma_{0,j}^2 = 1$ we ensure that, a priori, the processes $z_{t,j}$ are *stationary*, with $\boldsymbol{C}_t^{(j)} = \boldsymbol{C}_\infty^{(j)} = \frac{1}{2}\boldsymbol{I}$, i.e. no a-priori bias is introduced with regards to their transient characteristics.

We finally note that the autocovariance $\boldsymbol{D}_\tau^{(j)}$ of each of these stationary processes $j$ is given by:

$$
\boldsymbol{D}_\tau^{(j)} = \mathbb{E}\left[ \begin{bmatrix} \Re(z_{t+\tau,j}) \\ \Im(z_{t+\tau,j}) \end{bmatrix} [\Re(z_{t,j})\, \Im(z_{t,j})] \right] = \boldsymbol{A}_j^\tau \mathbb{E}\left[ \begin{bmatrix} \Re(z_{t,j}) \\ \Im(z_{t,j}) \end{bmatrix} [\Re(z_{t,j})\, \Im(z_{t,j})] \right] = \boldsymbol{A}_j^\tau \boldsymbol{C}_t^{(j)} \tag{26}
$$

where $\boldsymbol{A}_j$ and the covariance $\boldsymbol{C}_t^{(j)}$ are given above. By exploiting the diagonalization of $\boldsymbol{A}_j$ and that $\boldsymbol{C}_t^{(j)} = \boldsymbol{C}_\infty^{(j)} = \frac{1}{2}\boldsymbol{I}$ (for the parameter values discussed earlier), we obtain that:

$$
\boldsymbol{D}_\tau^{(j)} = e^{\tau\Re(\lambda_j)} \begin{bmatrix} \cos(\tau\Im(\lambda_j)) & -\sin(\tau\Im(\lambda_j)) \\ \sin(\tau\Im(\lambda_j)) & \cos(\tau\Im(\lambda_j)) \end{bmatrix} \tag{27}
$$

One can clearly observe harmonic (cross-)correlation terms which depend on the imaginary part of the $\lambda_j$ and can capture persistent periodic effects of the dynamical system in the long-time range.

## C   DERIVATION OF THE ELBO

This section contains details of the derivation of the Evidence-Lower-Bound (ELBO) which serves as the objective function for the determination of the parameters $\phi$ and $\theta$ during training. In particular:

$$
\begin{aligned}
&\log p(\boldsymbol{x}_{0:T}^{(1:n)}|\boldsymbol{\theta}) \\
&= \log \int p(\boldsymbol{x}_{0:T}^{(1:n)}, \boldsymbol{X}_{0:T}^{(1:n)}, \boldsymbol{z}_{0:T}^{(1:n)}, \boldsymbol{\theta}) \, d\boldsymbol{X}_{0:T}^{(1:n)} \, \boldsymbol{z}_{0:T}^{(1:n)} \\
&= \log \int \frac{p(\boldsymbol{x}_{0:T}^{(1:n)}|\boldsymbol{X}_{0:T}^{(1:n)}, \boldsymbol{z}_{0:T}^{(1:n)}, \boldsymbol{\theta})p(\boldsymbol{X}_{0:T}^{(1:n)}, \boldsymbol{z}_{0:T}^{(1:n)}, \boldsymbol{\theta})}{q_{\phi}(\boldsymbol{X}_{0:T}^{(1:n)}, \boldsymbol{z}_{0:T}^{(1:n)})} q_{\phi}(\boldsymbol{X}_{0:T}^{(1:n)}, \boldsymbol{z}_{0:T}^{(1:n)}) \, d\boldsymbol{X}_{0:T}^{(1:n)} \, d\boldsymbol{z}_{0:T}^{(1:n)} \\
&\geq \int \log \frac{p(\boldsymbol{x}_{0:T}^{(1:n)}|\boldsymbol{X}_{0:T}^{(1:n)}, \boldsymbol{z}_{0:T}^{(1:n)}, \boldsymbol{\theta})p(\boldsymbol{X}_{0:T}^{(1:n)}, \boldsymbol{z}_{0:T}^{(1:n)}, \boldsymbol{\theta})}{q_{\phi}(\boldsymbol{X}_{0:T}^{(1:n)}, \boldsymbol{z}_{0:T}^{(1:n)})} q_{\phi}(\boldsymbol{X}_{0:T}^{(1:n)}, \boldsymbol{z}_{0:T}^{(1:n)}) \, d\boldsymbol{X}_{0:T}^{(1:n)} \, d\boldsymbol{z}_{0:T}^{(1:n)} \\
&= \mathcal{F}(q_{\phi}(\boldsymbol{X}_{0:T}^{(1:n)}, \boldsymbol{z}_{0:T}^{(1:n)}), \boldsymbol{\theta})
\end{aligned}
\tag{28}
$$

# D    DETAILS FOR EXPERIMENTS

This appendix contains details for our experiments involving moving particles that have stochastic interactions corresponding to either an Advection-Diffusion behaviour or viscous Burgers' type behaviour.

## D.1    PARTICLE DYNAMICS: ADVECTION-DIFFUSION

For the simulations presented $f = 250 \times 10^3$ particles were used, which, at each microscopic time step $\delta t = 2.5 \times 10^{-3}$ performed random, non-interacting, jumps of size $\delta s = \frac{1}{640}$, either to the left with probability $p_{left} = 0.1875$ or to the right with probability $p_{right} = 0.2125$. The positions were restricted in $[-1, 1]$ with periodic boundary conditions. It is well-known (Cottet & Koumoutsakos, 2000) that in the limit (i.e. $f \to \infty$) the particle density $\rho(s, t)$ can be modeled with an advection-diffusion PDE with diffusion constant $D = (p_{left} + p_{right})\frac{\delta s^2}{2\delta t}$ and velocity $v = (p_{right} - p_{left})\frac{\delta s}{\delta t}$:

$$\frac{\partial \rho}{\partial t} + v\frac{\partial \rho}{\partial s} = D\frac{\partial^2 \rho}{\partial s^2}, \qquad s \in (-1, 1).. \tag{29}$$

From this simulation every 800th microscopic time step the particle positions were extracted and used as training data for our system. Sample initial conditions are shown in Figure 10:

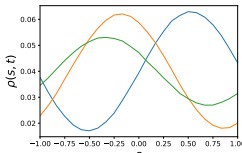

Figure 10: Sample initial conditions for the advection-diffusion type dynamics.

The architecture of the neural networks for the generative mappings described above as well as for the variational posteriors introduced in Section 2.2 can be seen in Figure 11. The neural network used for the generative mapping between the low-dimensional states $z_t$ and the mean and covariance for $X_t$ consists of only one dense layer, whereas the variational posterior on $z_{0:T}$ is parameterized using a dense Layer with ReLu activation followed by another dense layer.

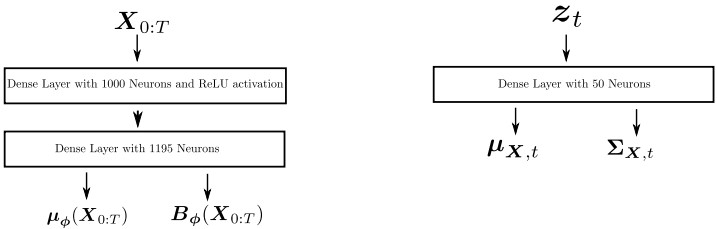

Figure 11: Neural Net architecture used for the particle dynamics corresponding to an advection-diffusion equation.

## D.2    PARTICLE DYNAMICS: VISCOUS BURGERS' EQUATION

The second test-case involved a fine-grained system of $f = 500 \times 10^3$ particles which perform *interactive* random walks i.e. the jump performed at each fine-scale time-step $\delta t = 2.5 \times 10^{-3}$ depends on the positions of the other walkers. In particular we adopted interactions as described in Roberts (1989); Chertock & Levy (2001); Li et al. (2007) so as, in the limit (i.e. when $f \to \infty, \delta t \to 0, \delta s \to 0$), the particle density $\rho(s, t)$ follows a viscous Burgers' equation with $\nu = 0.0005$:

$$\frac{\partial \rho}{\partial t} + \frac{1}{2}\frac{\partial \rho^2}{\partial s} = \nu\frac{\partial^2 \rho}{\partial t^2}, \qquad s \in (-1, 1). \tag{30}$$

From this simulation every 800th microscopic time step the particle positions were extracted and used as training data for our system. Sample initial conditions are shown in Figure 12.

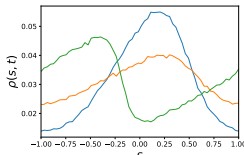

Figure 12: Sample initial conditions for the Burger's type dynamics.

The architecture of the neural networks for the generative mappings described above as well as for the variational posteriors introduced in Section 2.2 can be seen in Figure 13. The neural network used for the generative mapping between the low-dimensional states $z_t$ and the mean and covariance for $X_t$ consists of several dense layers with ReLu activation and Dropout layers to avoid overfitting, whereas the variational posterior on $z_{0:T}$ is parameterized using two dense layers with ReLu activation followed by another dense layer.

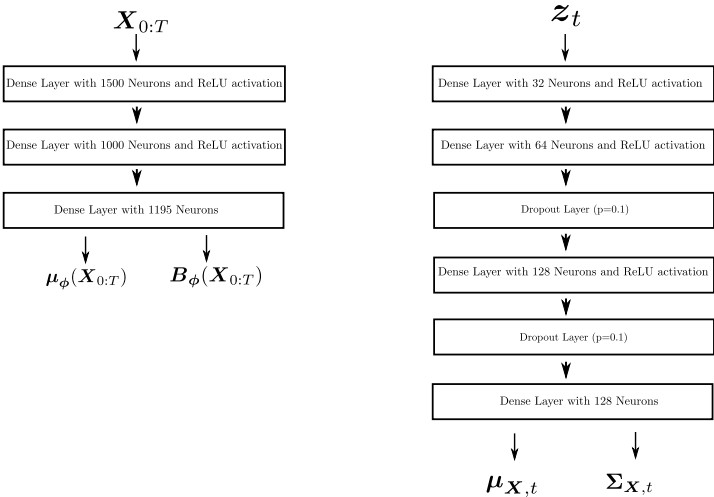

Figure 13: Neural Net architecture used for the particle dynamics corresponding to a viscous Burgers' equation.

# E  DETAILED ANALYSIS OF THE GENERATIVE MAPPING AND THE SLOW LATENT VARIABLES

In this Appendix we take a closer look at the generative mapping and the (slow) latent variables $z$ learned . For the Advection-Diffusion example, we discovered two slow processes (Section 4.1), $z_1$ and the marginally faster process $z_2$. The rest of the processes were very fast in comparison and took values close to the zero point of the complex plane during the inference as well as during the prediction phase.

In order to visualize the influence through the generative mapping of these two slow processes, we set the value of all other processes to zero and then reconstructed the fine-grained state based on different absolute values of $z_1$ and $z_2$. The result are shown in Figure 14 in terms of the reconstructed particle density. It is clearly visible that those two processes are responsible for a variety of density profiles. In accordance with their slowness, $z_1$ (the slightly slower process) is responsible for the most striking changes, whereas the other slow process generates some smaller scale fluctuations.

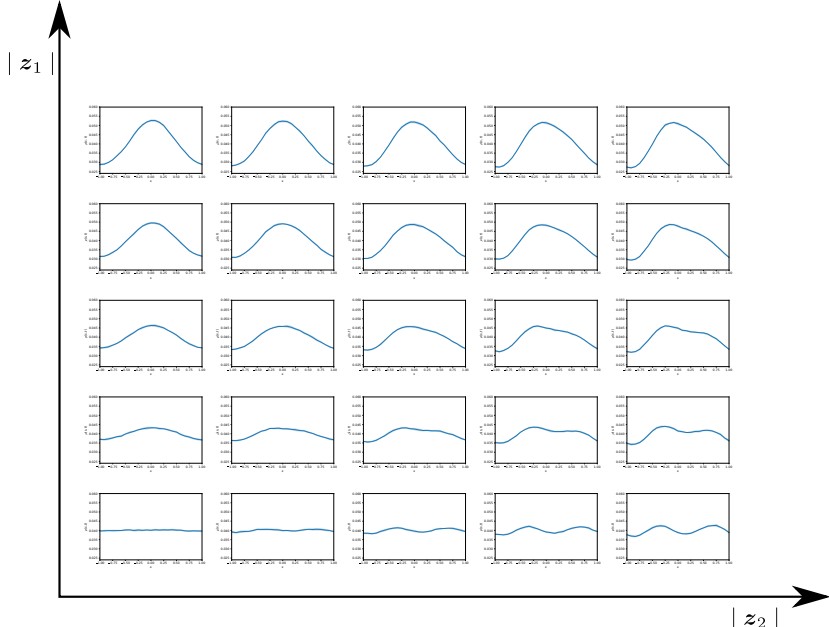

Figure 14: Reconstruction of the particle density profiles for various values of the two slowest processes $z_1$ and $z_2$ identified. All other latent variables $z_{t,j}$ were set to zero.

# F    Two-point Probability

This appendix contains the predictive estimates for the two-point probability, i.e. the probability of finding two particles simultaneously in two bins $(b_1, b_2)$. This two-point probability can be computed based on the reconstructed fine-grained system and corresponds to a second order statistic.

## F.1    Particle Dynamics: Advection-Diffusion

The estimated two-point probability as well as the comparison to test data is shown for two indicative time-instants in Figure 15 and 16. We note a very good agreement with the ground truth.

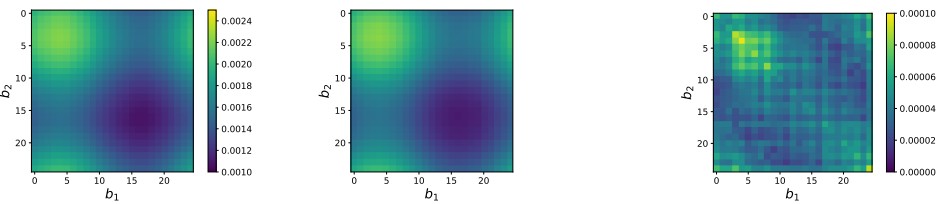

Figure 15: Two-point probability at time step 90: On the left the two-point probability of the data is shown as reference, the figure in the middle contains the predictive posterior mean whereas the figure on the right contains the standard deviation. The figure on the left and the figure in the middle share the same colorbar.



Figure 16: Two-point probability at time step 140: On the left the two-point probability of the data is shown as reference, the figure in the middle contains the predictive posterior mean whereas the figure on the right contains the standard deviation. The figure on the left and the figure in the middle share the same colorbar.

## F.2    Particle Dynamics: Viscous Burgers' equation

The estimated two-point probability as well as the comparison to test data is shown for two indicative time-instants in Figure 17 and 18. We note a very good agreement with the ground truth.



Figure 17: Two-point probability at time step 90: On the left the two-point probability of the data is shown as reference, the figure in the middle contains the predictive posterior mean whereas the figure on the right contains the standard deviation. The figure on the left and the figure in the middle share the same colorbar.



Figure 18: Two-point probability at time step 140: On the left the two-point probability of the data is shown as reference, the figure in the middle contains the predictive posterior mean whereas the figure on the right contains the standard deviation. The figure on the left and the figure in the middle share the same colorbar.

# G EFFECT OF THE AMOUNT OF TRAINING DATA

This section contains a study on the influence of the amount of training data. We illustrate this in the context of the Advection-Diffusion example (section 4.1) by using 16 time sequences instead of the 64 employed earlier. In the figures below we note that the proposed model is capable capturing the main features of the system's dynamics as well as the correct steady state even with fewer training data. We believe that this is due to the intermediate layer of physically-motivated variables $X_t$ which introduces an information bottleneck. Finally, and as one would expect, fewer training data leads to increased predictive uncertainty.

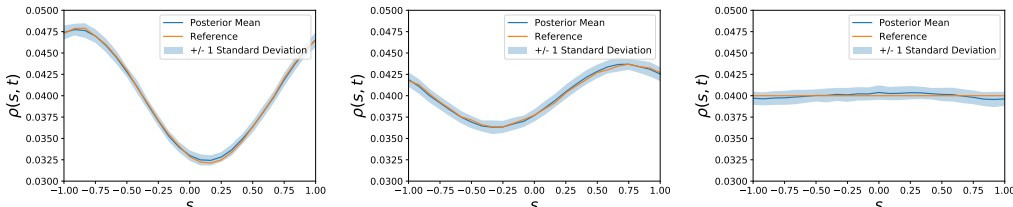

Figure 19: Predicted particle density profiles at $t = 80, 160, 1000$ (from left to right) with 64 samples.

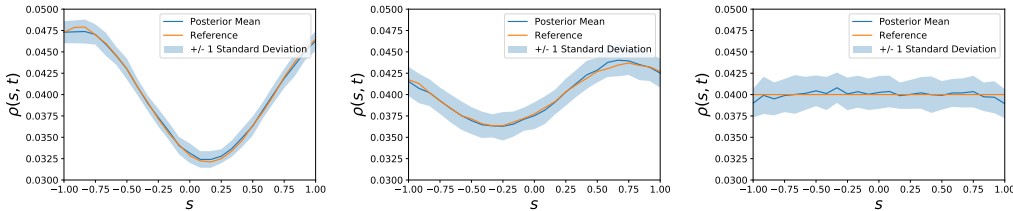

Figure 20: Predicted particle density profiles at $t = 80, 160, 1000$ (from left to right) with 16 samples.

# H    COMPARISON WITH OTHER APPROACHES

This appendix contains results obtained by other methods for the test cases discussed in the main text. The simulation data is identical to the one used for the proposed method and details of the specific algorithms are described in the following.

## H.1    REAL-VALUED LATENT SPACE

To demonstrate the utility of a complex-valued latent space, the two examples were also solved with a real-valued $z_{t,j}$. The only difference here is the restriction of the latent variables $z_{t,j}$ and the $\lambda_j$ to real values.

A model with real-valued latent space is also capable of ensuring the stability but it is not capable of capturing periodic components of the dynamics. As most physical systems (including the two examples) contain such components, the algorithm is not able to accurately model the dynamics and fails in generating reliable predictions. This can be readily observed in the extrapolative predictions of Figures 21 and 22 where, apart for the biased results, one can also note increased predictive uncertainty.

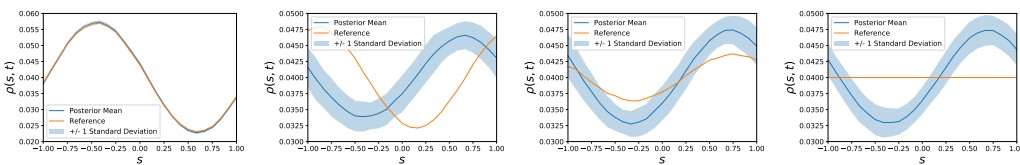

Figure 21: Advection-Diffusion system: Predictions at $t = 0, 80, 160$ and $1000$ obtained with real-valued latent variables $z_{t,j}$.

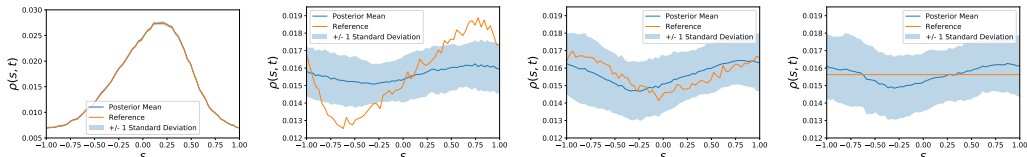

Figure 22: Burgers' system: Predictions at $t = 0, 80, 160$ and $1000$ obtained with real-valued latent variables $z_{t,j}$.

## H.2 No stable latent space

Another possibility is to employ only the physically motivated latent variables $\boldsymbol{X}_t$ and remove completely the latent variables $\boldsymbol{z}_t$ in the first layer. This approach is similar to the one investigated in Felsberger & Koutsourelakis (2019) and Kaltenbach & Koutsourelakis (2020) as well as to the idea of neural ODEs (Chen et al., 2018). In this case, one must learn directly the dynamics of $\boldsymbol{X}_t$ and for this purpose we employed a three-layer, fully-connected neural network $NN$ as follows:

$$\boldsymbol{X}_{t+1} = NN(\boldsymbol{X}_t) + \sigma\boldsymbol{\epsilon}, \qquad \boldsymbol{\epsilon} \sim \mathcal{N}(\boldsymbol{0}, \boldsymbol{I}). \tag{31}$$

The learned dynamics are in general non-linear and stability is not guaranteed. As it can be observed in Figures 23 and 24, the trained model is capable of producing accurate predictions for some time-steps but eventually in both cases predictions become unstable. This could also be problematic when the trained model is used to make predictions with new initial conditions as the chaotic nature of the nonlinear dynamics can lead to significant errors even for shorter time horizons.

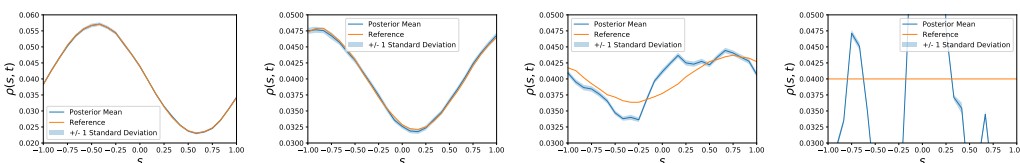

Figure 23: Advection-Diffusion system: Predictions at $t = 0, 80, 160$ and $1000$ obtained without $\boldsymbol{z}_t$ and with the model of Equation (31).

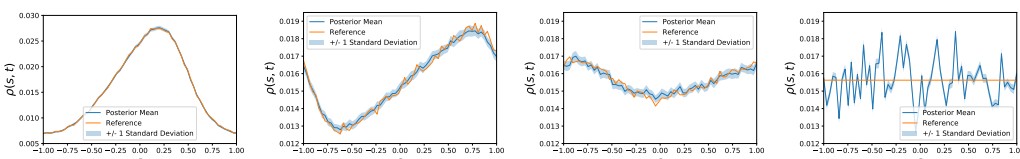

Figure 24: Burgers' system: Predictions at $t = 0, 80, 160$ and $1000$ obtained without $\boldsymbol{z}_t$ and with the model of Equation (31).

### H.3 KOOPMAN-BASED MODELS

The final alternative explored involved probabilistic and deterministic Koopman-based models for the latent dynamics. We kept the generative framework of our model and did not use an encoder as for instance in Gin et al. (2019) in order to remove the effect of the associated model choice. For the same reason, we retained the intermediate variables $X_t$ even though these do not appear in any known Koopman-operator implementations. We replaced our complex-valued dynamics of the latent processes $z_{t,j}$ with the models described in the sequel.

#### H.3.1 PROBABILISTIC KOOPMAN-BASED MODEL

We used real-valued latent variables $z_t$ which are not a-priori independent and whose dynamics are parameterized with a Koopman matrix $K$ and a diagonal noise matrix $W$:

$$z_{t+1} = K z_t + W \epsilon, \qquad \epsilon \sim \mathcal{N}(0, I). \tag{32}$$

The learned matrix $K$ is not guaranteed to be stable in the absence of additional constraints but in both cases examined the eigenvalues of the learned $K$ were real smaller than one. Long-term predictions were stable and for the Burgers' case in Figure 26 we were also able to reach the true steady state. For the (simpler) Advection-Diffusion example in Figure 25 an incorrect steady state was reached and the predictive quality started to deteriorate after some time-steps. In comparison to our framework, the probabilistic Koopman-based model does not provide a direct separation between slow and fast processes and therefore its interpretability is reduced.

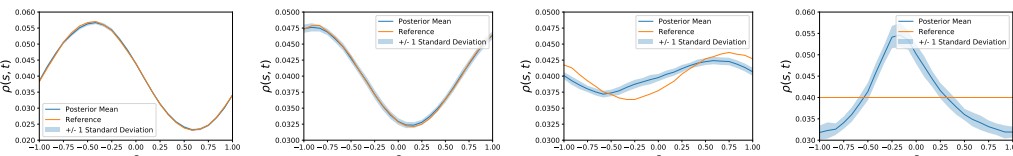

Figure 25: Advection-Diffusion system: Predictions at $t = 0, 80, 160$ and $1000$ obtained with the probabilistic Koopman-based model of Equation (32)).

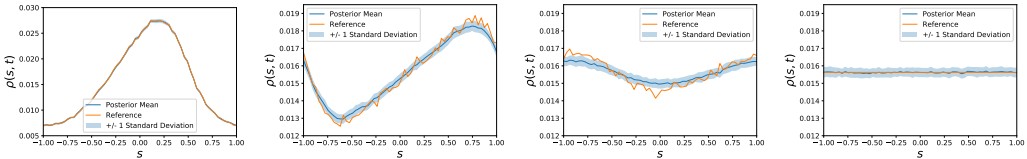

Figure 26: Burgers' system: Predictions at $t = 0, 80, 160$ and $1000$ obtained with the probabilistic Koopman-based model of Equation (32)).

#### H.3.2 NON-PROBABILISTIC KOOPMAN LEARNING

We also used real-valued latent variables $z_t$ with deterministic dynamics which were parameterized as follows:

$$z_{t+1} = K z_t \tag{33}$$

The absence of noise in comparison to Equation (32), led in both cases to an estimate for the Koopman matrix $K$ that did not yield stable predictions (each of the learned $K$ matrices had at least one eigenvalue which was larger than 1). We speculate that the lack of stochasticity made the model less capable of dealing with the information loss. We also note in Figures 27 and 28 that the predictions obtained are, with an exception of a few time-steps, highly inaccurate.

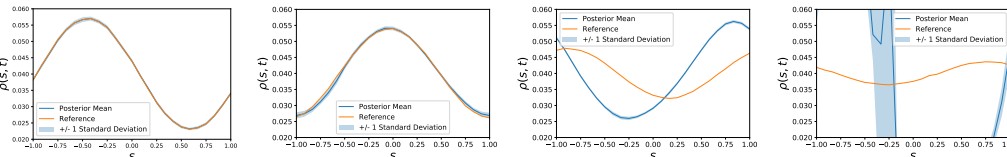

Figure 27: Advection-Diffusion system: Predictions at $t = 0, 20, 80$ and $160$ obtained with the deterministic Koopman-based model of Equation (33)).

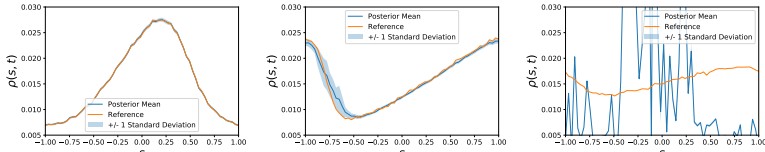

Figure 28: Burgers' system: Predictions at $t = 0, 20$ and $80$ obtained with the deterministic Koopman-based model of Equation (33)).

