# OpenReview forum: "Physics-aware, probabilistic model order reduction with guaranteed stability"
_ICLR.cc/2021/Conference — ICLR 2021 Poster_

### Official Review · AnonReviewer4 · 2020-10-23
**Interesting paper that adds a physics-motivated latent space to the standard latent space**

**Rating:** 7
**Confidence:** 4

**Review:**

The paper proposes a generative model for learning a low-dimensional representation of a dynamical system from high-dimensional observations. The novelty of the approach is to introduce two latent spaces, one representing the standard physics-agnostic latent space learned from the data and one representing physics-motivated variables. The goal is to learn the dynamics of the first layer, denoted by z_t, which is a coarse-grained representation of the dynamical system and mostly captures the slow processes that drive the system.

Quality, clarity, originality and significance:
Pro: The paper is very well written, clear, and the generative approach is novel. Adding domain knowledge is relevant and significant when dealing with real world applications. Cons: Below are a few comments and questions to clarify some of the aspects and choices made.

How are the physics-motivated variables chosen? Is there a guarantee that the chosen representation/variables are sufficient to allow a one-on-one mapping between the physics-based latent space $X_t$ and the observations/data $x_t$? (As a small comment, I found it slightly confusing to use the same letter for the latent space and data space, even if one is lower case and the other upper case.)

What does equilibrium represent here? Does it necessarily need to be a stationary process as in the experiments, or can it be a trend, or a periodic signal? Are the latter two cases handled well by the method?

Fig. 1: Would it be possible to add the maps F and G to the figure? It is not very clear to me how the latent variables X are obtained, are they computed from the data? If X is the generative process for x, why do we need the latent space z? Would it be possible that the latent space X is sufficient, and if not, would the authors have examples when this is not the case?

The framework is applied to physics simulations. How easy/difficult would it be to apply the method to real world data, and what would be the main challenges?

Sect. 2: Should it be $x_t \in \mathcal{M}$, instead of $x \in \mathcal{M}$? The paper mentions that the $z_t$’s correspond to a nonlinear coordinate transformation, but do not specify of what?

What is the motivation for using the Ornstein-Uhlenbeck process for z, and what would be other alternatives? Why exactly does z need to be complex, is it in order to be able to capture slow and fast processes? In Figs. 2 and 5, the slow processes seem to have imaginary part close to zero, and fast processes real part close to zero. What is the explanation for this (this might be a known fact in dynamical systems theory, but not so much in ML)? Does it always have to be the case that the \lambda’s have either the real or imaginary parts close to zero? What would happen if the system is still multi-scale but on a much more continuous scale than in the experiments presented here and where there wouldn’t be such a clear distinction between slow/intermediate/fast scales (this would probably be the case in complex real world systems)?

According to eq (4) to ensure long-term stability, the real part of $\lambda_j$ should be
negative. This only applies to the slow processes, is that right? What is an explanation for this?

In the definition for $\sigma_j$ there is a 2 factor which is not in eq (7). Same line: the latent dynamics are stationary refers to which layer, $z_t$ or $X_t$? Using clearly defined names for the two latent spaces would make the paper easier to read. Maybe even add this to Fig. 1.

In ML we don’t see often complex distribution, and it would be good to specify that $\mathcal{CN}$ is the complex distribution.

Eq (1): as $\lambda$ and $\sigma$ do not depend on time, it means that all hidden processes $z_t$ follow the same dynamical model. What are the implications of this, and would this be a reasonable choice in a real world scenario?

How robust are the results to the choice of the number of latent variables h, and the choice of the physics-based variables?

--------------------------
Rebuttal: Thank you to the authors for their detailed response. I am happy with the response and will keep the original score.

---

> ### Author Response · Authors · 2020-11-18
> **Response to Reviewer 4**
>
> We thank the reviewer for their assessment and the positive feedback as well as the suggestions for improving the paper.
>
>
> - The physics-motivated variables have to be chosen a priori and necessitate the availability of  some physical knowledge about the system. A one-to-one mapping will (almost) never  exist as long as the physically-motivated variables $\boldsymbol{X}$ (or more generally the reduced description) are lower-dimensional than the observables $\boldsymbol{x}$. The goal of predictability can be achieved if the conditional of $\boldsymbol{x}$ given $\boldsymbol{X}$ is unimodal and preferably of low variance.
> As we pointed in  our response to reviewer R1,  we have investigated  cases where the $\boldsymbol{X}$ used was known to provide an incomplete picture. Our  preliminary  observations suggested an increased uncertainty in the reconstructed $\boldsymbol{x}_t$ i.e. insufficient $\boldsymbol{X}_t$ implied increased stochasticity in the predicted $\boldsymbol{x}_t$.
> In a way, this is already the case in the examples considered, i.e. even if one assumes that the particle density is sufficient for the reconstruction of the particle positions $\boldsymbol{x}_t$, the $\boldsymbol{X}$ that we employ are based on a discretization of this density which unavoidably implies some information loss.
> Nevertheless, for a lot of applications or observables,  increased predictive uncertainty  could constitute predictions irrelevant or impractical and  reduce the utility of the method proposed. An option that we also mention in the Conclusions is to include (some of) the $\boldsymbol{z}_t$ in the decoder. We note though that  increasing the inputs of the decoder, complicates the associated map and unavoidably would increase the amount of training data needed. An option we are exploring is to use only  a small number of features of $\boldsymbol{z}_t$ which are identified by sparsity-enforcing priors.  We would be happy to expand on the technical aspects, if the reviewer requests it. As  a side note regarding the notational conventions, and while we do not claim that our choices were optimal for a diverse audience such as ICLR's, we would like to explain our reasoning for $\boldsymbol{X}_t$: Given the notation $\boldsymbol{x}_t$ for the fine-scale observables,  we selected a capital letter as it alludes to a macro/coarse-description and the same letter as for the fine-scale system  to evoke  a physical meaning.
>
>
>
> - The notion of equilibrium is meant in the statistical sense i.e. the density of particle positions' $\boldsymbol{x}$ becomes independent of time $t$. This does not mean that the system cannot change in time, but rather that its  statistics would remain  constant. In our formulation, transient characteristics are determined by the latent processes $\boldsymbol{z}_{t,j}$. We have added some details in Appendix B regarding their  stationary density as well as their autocovariance. We show that the latter contains modulated periodic functions that depend on the imaginary part of the learned parameters $\lambda_j$. We finally note that due to the nonlinear maps involved, the resulting equilibrium density of $\boldsymbol{X}$ and more importantly of $\boldsymbol{x}$, will be highly non-Gaussian.
>
> - We improved Figure 1 according to the reviewer's suggestions. We note that the latent, physically-motivated variables $\boldsymbol{X}$ act as an information bottleneck and facilitate the reconstruction of the high-dimensional observables $\boldsymbol{x}$. The dynamics of $\boldsymbol{X}$ can be highly nonlinear and non-Markovian. As we demonstrate in the newly added Appendix H.2, learning their dynamics from data does not guarantee long-term stability. Hence, the latent variables $\boldsymbol{z}$ are employed in an additional layer in order to capture their dynamics in a manner that always guarantees stability, but also provides interpretability by ordering processes based on their slowness.
>
> - The most important challenge for the application of the proposed model to other  dynamical systems pertains to the definition of the physically-motivated variables $\boldsymbol{X}$. These constitute the necessary information bottleneck that enables us to model  very high-dimensional systems with small amounts of training  data. While such variables are usually readily available in large classes of physical problems, this might not be the case in other contexts, e.g. financial time-series.
>
>
> - We thank the reviewer for pointing out the mistake in the notation of $\boldsymbol{x}_t$.

---

> > ### Author Response · Authors · 2020-11-18
> > **Response to Reviewer 4 [continued]**
> >
> > - The motivation behind the modeling assumptions for  $z_{t,j}$ is multi-fold. Firstly, the dynamics selected (i.e. $AR(1)$ or discretized  Ornstein-Uhlenbeck processes) guarantee long-term stability for all possible parameter values. Secondly, they enable an interpretable disentanglement of the underlying dynamics on the basis of their slowness. Capturing the slowest evolving features is especially important in multiscale problems as these determine the long-term evolution of the system. Finally, and while this is not explored in the paper, the simplicity of these processes would allows to readily consider continuous time versions. These would be able to handle observables at non-regular time-intervals and even invalidate the need of defining a time-step.
> > The novel contribution of our paper in this regard consists of the extension of this processes in the complex domain. As discussed  in the expanded Appendix B, this enables us to capture harmonic effects that frequently are present in dynamics of physical systems.
> > The reviewer is right with regards to the necessary condition for negative real-parts for the parameters $\lambda_j$. We note that in Figures 2 and 7 the horizontal axis corresponds to the real-part, and the vertical to the imaginary (we have improved the labels of these Figures). The former represents the rate of time decay whereas the latter the harmonic parts. The lack of clear distinction between slow and fast scales would not constitute a big problem. More important is in our opinion the number of latent processes $z_{t,j}$ that are needed to capture these scales. A continuous spectrum of equally-important time scales would most probably necessitate the use of several $z_{t,j}$.
> > An obvious deficiency of the proposed model is the lack of an automated procedure for determining the appropriate number of $\boldsymbol{z}$.
> > We believe that the ELBO which provides a lower bound on the model evidence could be used to that end, but some technical aspects would need to be resolved first.
> >
> >
> > - We have addressed and clarified all issues related to $\sigma_j$ and their effect. In Appendix B we provide further justification for the selected values of $\sigma_j$.
> >
> > - We have  added a better explanation for the complex-normal density  in the footnote as well as in Appendix A.
> >
> > - While the parameters $\lambda_j$ and $\sigma_j$ are independent of time $t$, they generally differ with $j$ i.e. each latent process $z_{t,j}$ has distinct dynamics.
> >
> >
> > - With regards to the number of $z_{t,j}$ variables, we have noted in our experiments that additional ones are associated with very fast dynamics and capture stochastic small-scale fluctuations. While these become irrelevant in long-range predictions, they might be important for some observables (higher-order statistics in particular) and for (very) short-term predictions.
> >
> >
> > - With regards to the physics-based variables $\boldsymbol{X}$ and as mentioned in responses to other reviewers as well, these   necessitate the availability of  some physical knowledge about the system.
> > They constitute the necessary information bottleneck that enables us to model  very high-dimensional systems with small amounts of training  data. While such variables are usually readily available in large classes of physical problems, we recognize that this might not be the case in other contexts, e.g. financial time-series.

---

### Official Review · AnonReviewer3 · 2020-10-27
**Good ideas - weak applications - several points to be improved**

**Rating:** 7
**Confidence:** 4

**Review:**


This paper proposes a new dimensional reduction scheme, based on a latent representation, using a generative model and variational inference.

The method is successfully applied to 2 simple one-dimensional physical processes, for which the macroscopic equations are known from the physics literature.

Choosing only a couple of simple physical variables X (as far as I understand, here it's the density in binned areas of the domain), and an arbitrary number (5) of latent variables (here we see that 5 is enough because it captures all the slow degrees of freedom), the model is able to learn the slow-variables values (eigenvalues controlling the evolution of the macroscopic variables) and then correctly reproduce the long-time behavior of the macroscopic observables (the X), and additionally produce realistic noise (x) / realistic microscopic fluctuations (this last part is not very clear, I am guessing here).
In addition, the latent representation of previously unseen initial conditions can be inferred (with decent error bars), allowing to predict the future of unseen initial trajectories.


Overall, the idea is interesting and supported by correct mathematical derivations and experimental proofs of concept.

Thus I lean on accepting the paper.


However, I have some questions and remarks that I would like to be addressed.


My main criticism would be the following: the paper does not use any pre-existing benchmarks, which makes accuracy comparisons essentially impossible.
Of course one may object that the novelty of the approach makes comparison impossible, however I think that it should be possible to re-implement State of the art methods (by the way I appreciated the literature review section), and show how much the new method deals well with smallish data sets (which I believe it does ?).

My second main criticism is that although the method is general and in principle applicable to complex problems, here it is applied to textbook cases, for which the exact macroscopic equations are known, and for which a mechanically stable state is reached in finite time, in other words, two very simple problems. (although if I recall well the equations (26), (27) do not allow a simple diagonal form in the style of (21)).
The simplicity of the problems attacked is seen also in the important (but hidden) remark, that equation (25) is sufficient to produce x, and that "no parameters need to be learned for p(x_t|X_t)".
What would happen for more complicated problems ? (e.g. protein folding, even in the case of small molecules like butane (C2H6 if I recall is a simple standard)? )
In particular, if the true number of "independent" components (number of lambda's) needed was very large ?


Also I have a trivial question, that however I think needs to be discussed in the paper. What is the error on the prediction of the microscopic positions of the {x} variables (the elementary particles) ? My guess is that by construction, the generative model has a 100% error (as good as random, I mean), because it does not track the future of individual particles, but rather predicts the future of the large-scale, slowly-evolving coarse-grained variables (X, governed by the latent z's).
Am I correct ? I think it may be worth mentioning that quickly, for the readers that are unfamiliar with the topic.
If I am completely wrong, it is then even more worth mentioning the accuracy of x(t+P) predictions.

I did not understand why section H of appendix was not part of the main text. It seems quite central to be able to forecast unseen initial conditions, and not just the future of already seen trajectories.




Less important remarks about how the paper presents the research:

sec 2.2 : additional intuitive explanation about how the learning is performed would be welcomed.
It is not clear to the unfamiliar reader as to how learning proceeds.
It seems to me that you directly learned the parameters of the joint probability distribution, which is quite factored, as you showed in Eq. (8), but needs Variational Inference methods to be solved for. What I don't see clearly is the shape of q_\phi, or rather, what it means for B to be bi-diagonal. Maybe you could explain which variables are connected with which, given your assumptions on B_\phi.
Also, after reading Appendix , Figure 9 and 11 could be compressed and included in the main text, or at least referred to explicitly.

A number of graphs are un-readable, in particular the x- and y-axis labels are so tiny that one cannot get what is plotted against what.
Please correct that, increasing both figure size and x/y-labels sizes.


-------
a couple of detailed remarks:

In Fig 13-16, the central plot does not have a colorbar. Is it because it's exactly the same as for the left plot ? If so, mention it or center the colorbar in a visually suggestive way

typo: "is is" (search it)

---

> ### Author Response · Authors · 2020-11-18
> **Response to Reviewer 3**
>
> We thank the reviewer for their thorough  analysis and  positive appraisal as well as the constructive criticism .
>
>
> - We have  added a new section (Appendix H) where we evaluated the performance of various approaches on our two test examples. These include deterministic and probabilistic models such as those based on  the Koopman operator or those employing deep neural networks to parameterize the latent dynamics. We believe that the results obtained support our modeling choices in terms of the complex-valued latent variables that capture slow-varying transients and enforce long-term stability, the use of the intermediate, physically-motivated variables as well as the adoption of a probabilistic framework. As our approach does not require  time-derivatives of the observables (which we consider a significant advantage as in most applications derivatives are estimated with finite differences and are therefore noisy), we did not compare our method with algorithms which do require such derivatives. We would also like to note that even though a wide variety of methods for the analysis of time-series (irrespective of their origin) have been proposed, their applicability would be limited or even impossible due to the very large number of observables (i.e. particles) in the problems considered.
>
> - The reviewer correctly points out that in our example the mapping $p(\boldsymbol{x}_t | \boldsymbol{X}_t)$  is predetermined. This was previously contained in the appendix but in the updated version has been moved to the main text.
> Furthermore, the reviewer is also right that the viscous Burgers' equation upon discretization does not lead to linear dynamics and special care has to be taken to ensure that solution does not "blow up" i.e. to ensure stability. Interestingly for such a nonlinear PDE, it is known that there is a nonlinear transformation (the Hopf-Cole transform) that converts it to the well-behaved linear heat equation. This is effectively the role of the latent variables $\boldsymbol{z}$ and the associated transformation $G$ in Equation 2, i.e. to learn nonlinear transformations that constitute the $\boldsymbol{X}$-dynamics linear (and Markovian). Furthermore, even if the Cole-Hopf transfrom was a-priori enforced, the spatial-discretization of the new field would require more degrees of freedom than the $5$ complex-valued variables $\boldsymbol{z}$ with which our model is able to encode the system's dynamics.
> Finally with regards to other examples, we recognize that the central role of the physically-motivated variables $\boldsymbol{X}$ precludes the direct application in problems (like the single molecules mentioned) where such variables are not readily available or some dimensionality reduction technique would have to be applied beforehand in order to identify them. For low-dimensional systems, the utility of our method would be reduced as in such cases there is generally no need to  find lower-dimensional descriptions. With regards to a high number of $\lambda$'s needed, we note that an automatic selection of the right dimension of the low-dimensional state-space is a direction for future work as also pointed out by other reviewers. If the chosen latent variables are not capable of expressing the dynamics than the predictive uncertainty will become high.
>
>
> - The reviewer addresses a common misunderstanding for such (single-species) particle systems and gave us the opportunity to clarify this in the updated text. While the system of interest consists of a very large number of particles these are exchangeable or put differently, the system is permutation- invariant. One does not care about an individual particle but rather than first-, second- and higher-order interactions such as the ones reported in the text and in Appendix F. On one hand this simplifies the task but it is also well-known that enforcing a priori such invariances (or more generally symmetries, equivariances etc) is not easy particularly if highly-expressive models (such as neural networks) are employed. In part, the physically-motivated variables $\boldsymbol{X}$ are introduced to facilitate this task.
>
> - We agree with the reviewer that Appendix  H (with old numbering)  is very important and have now moved it to the main text.
>
> - We took again a closer look at our inference section and added more explanations regarding the factorization of our variational distributions. We also improved the graphs and the color-bar in Figures 13-16. We have also addressed all the editorial remarks   and are thankful to the reviewer for pointing them out.

---

### Official Review · AnonReviewer1 · 2020-10-29
**Nice way of combining physics**

**Rating:** 6
**Confidence:** 5

**Review:**

This paper presents a generative state-space model using two layers of latent variables. The latent variables in the first layer aim to capture long-term dynamics and ensures the stability. The variables in the second layer are physical variables. The authors have shown some promising results in modeling particle dynamics.

Since the authors claim that the use of X can reduce the complexity/the search space of the learning model, it would be great if the authors can show how the performance change given different number of training samples.

Another issue is about the effectiveness of this model in simulating real-world physical phenomena. This method can work well on simulated dataset because simulated data always follow the dependencies between X and x (as the simulator is built based on these rules). Real-world physical systems can be complex and usually we do not know exactly the governing physical variables (X). It would be great to discuss whether the proposed method allows some flexibility to automatically discover these unknown physical variables.

---

> ### Author Response · Authors · 2020-11-18
> **Response to Reviewer 1**
>
> We thank the reviewer for their comments and the positive feedback as well as for the suggestions for improving the paper.
>
> - We have included  a  study in  the newly-added Appendix G of the  paper that examines the effect of the amount of  training data. We have observed that even with a quarter of the original data, the model can capture the salient dynamical features and the correct steady state.   We believe that this is due to the intermediate layer of physically-motivated variables $\boldsymbol{X}_t$ which introduce an information bottleneck. As one would perhaps expect, fewer training data  leads to increased predictive  uncertainty. We further  contrast this performance with several other methods that have been added in Appendix H.
>
> - The reviewer is right that in a lot of physical systems we do not know exactly the governing physical variables. Very often some physical variables are available but they do not necessarily provide the full picture (e.g. unknown internal state variables). In such a discussion, we should distinguish  between unknown variables that are needed to capture the dynamics of (known variables or variables of interest) $\boldsymbol{X}_t$ and, secondly, unknown variables that are needed to reconstruct $\boldsymbol{x}_t$. For the first case, the premise of the method is that given a sufficient number of $\boldsymbol{z}_t$ , the dynamics of $\boldsymbol{X}_t$ can be correctly captured. This should apply to cases where memory terms appear in the latter i.e. we exchange memory for additional variables [Kondrashov et al. : Data-driven non-Markovian closure models, 2015]. We recognize that  an automated scheme for determining the right number of $\boldsymbol{z}_t$ is currently missing. The second case would require augmenting the inputs of the decoder $\boldsymbol{F}$ in Equation 3. We are currently investigating such cases and preliminary results (which do not constitute conclusive evidence) suggest an increased uncertainty in the reconstructed $\boldsymbol{x}_t$ i.e. insufficient $\boldsymbol{X}_t$ implied increased stochasticity in the predicted $\boldsymbol{x}_t$.
> In a way, this is already the case in the examples considered, i.e. even if one assumes that the particle density is sufficient for the reconstruction of particle variables $\boldsymbol{x}_t$, the $\boldsymbol{X}$ that we employ are based on a discretization of this density which unavoidably implies some information loss.
> Nevertheless, we recognize that for a lot of applications or observables,  increased predictive uncertainty  could constitute predictions irrelevant or impractical and  reduce the utility of the method proposed. An option that we also mention in the Conclusions is to include (some of) the $\boldsymbol{z}_t$ in the decoder. We note though that  increasing the inputs of the decoder complicates the associated map and unavoidably would increase the amount of training data needed. An option we are exploring is to use only  a small number of features of $\boldsymbol{z}_t$ which are identified by sparsity-enforcing priors.  We would be happy to expand on the technical aspects, if the reviewer requests it.

---

### Official Review · AnonReviewer2 · 2020-10-29
**Interesting paper. But some parts can be improved.**

**Rating:** 7
**Confidence:** 3

**Review:**

This is a good paper, in my opinion. I have some suggestion to improve the quality of the work.
See below.

 - I suggest to improve Section 3, open a bit the range of references,  considering general and relevant works such as

C. Grigo et al. A physics-aware, probabilistic machine learning framework for coarse-graining high-dimensional systems in the Small Data regime
arXiv:1902.03968, 2019.

G. Camps-Valls, et al. "Physics-Aware Gaussian Processes in Remote Sensing", Applied Soft Computing, Volume 28, Pages: 69-82, 2018.

Sungyong Seo et al. "Physics-aware Difference Graph Networks for Sparsely-Observed Dynamics",  ICLR 2020,

- Explain better figure 2, improving its caption.

- Figure 3 is completely unclear, remove it or improve.

- Please explain better your system in Eqs (1)-(2)-(3). What are the measurements/observations? what is your inference goal? please clearer state these points.

---

> ### Author Response · Authors · 2020-11-18
> **Response to Reviewer 2**
>
> We thank the reviewer for their comments and the positive feedback as well as for the suggestions for improving the paper.
>
>
> - We have incorporated the papers suggested into the discussion and thank the reviewer for pointing  them out.
>
> - The captions and explanations for Figures 2-3 as well as the Equations (1)-(3) have also been improved  and the  points mentioned are  hopefully more clearly presented. Additional clarifications regarding the latent dynamics of $z_{t,j}$ as well as the associated parameter choices have been added in Appendix B.

---

### Official Review · AnonReviewer5 · 2020-11-09
**Interesting idea with encouraging results but lacks necessary motivation and ablations (updated)**

**Rating:** 6
**Confidence:** 3

**Review:**

Summary:

The paper presents a generative approach to modeling physical systems with high-dimensional, nonlinear dynamical systems such as those found in fluid mechanics. The authors provide a physics-motivated hierarchical model for high-dimensional time series and a variational inference method for inferring latent variables and dynamical system parameters. They demonstrate its performance on simulated fluid mechanics prediction tasks.

Strong points:

The paper presents an interesting and motivating case for Bayesian inference in probabilistic generative models: a problem that has inherent uncertainty along with the ability to incorporate domain knowledge that can reduce the inference complexity.

The results on the simulated tasks are encouraging, especially the results in Appendix H which demonstrate generalization to out-of-distribution data. The paper does an excellent job of presenting and analyzing the results, along with compelling visualizations.

In general, the paper is well written (apart from some higher-level structural issues discussed below) and the notation is clear and unambiguous.

Weak points:

Structurally, the paper could do a better job of motivating the particular choices made when designing the model. The choices that could be made more clearly motivated are:

1) using a complex-valued latent dynamical system. Although there is a sentence in the “Related Work” section (“the prior proposed and the use of complex variables enable the discovery of slow features”), this important choice could use additional discussion and motivation.

2) On first read, it was unclear what the likelihood model was, especially when it was stated that the observed data were time-series of (e.g.) particle velocities and positions. In Appendix E, it is stated that a Multinomial observation model is used, which could not be used for particle velocities/positions since it models counts of discrete variables. Thus, the observations $x_t$ in this case are statistics of velocities/positions (i.e. counts of particles in discrete buckets). This choice should be discussed in the main text, as it both changes the dataset size and dimensionality, along with the goal of the model itself. Rather than modeling particle positions and velocities, it seems like the model is generating statistics of the system. While this isn’t inherently problematic, this choice is not clearly indicated in the text.

3) Finally, the paper includes $z_t$ in the model to help make predictions converge to equilibrium. While intuitively, having $z_t$ decay according to the model corresponds to some notion of “stability”, the paper should also explain how the exclusion of $z_t$ can lead to unstable or diverging predictions. Although the paper cites related work to justify the choice, it is such an important choice that I think it merits more discussion in the main text.

These structural issues with the paper are worsened by the fact that there are no ablations or comparisons in the paper. I think the minimal set of additional experiments would include 1) a model with real-valued (not complex) latent dynamics and 2) a model without $z$ in it, to demonstrate the necessity of a stabilizing element in the model. Even better would be comparisons to non-probabilistic models, demonstrating the necessity of uncertainty in the face of “information loss”. Finally, comparisons to related methods (e.g. Koopman-operator baselines) would help understand how the proposed model improves on the surrounding literature.

Recommendation:

While I find the problem setting and proposed approach very interesting, the writing of the paper and the results still need work. The paper needs to better justify the modeling choices both in writing and with additional experiments. The provided results are encouraging, but I think a proper evaluation of the approach with the appropriate ablations and baselines is necessary for publication. I recommend a reject on these grounds.

Clarifying questions:

Are predictions obtained using MAP estimates of the parameters $\theta$ or by sampling the posterior predictive distribution (i.e. marginalizing out $\theta$)?
In what way is your inference approach “hybrid” as opposed to fully Bayesian?

Additional feedback:

Defining terms like “fine-grained” and “multiscale” would help make the paper more accessible to readers without physics backgrounds
Nit: “parametrized” -> “parameterized”
Page 4: “algrithm” -> “algorithm”


===========================================
Updates:

After considering the author's response and updates to the paper, I have bumped the score to a 6. The addition of Appendix H, in my opinion, considerably strengthens the paper's story and case for acceptance. I still have minor concerns about the writing surrounding the use of the Multinomial likelihood - for example, the paper still claims to be dealing with high-dimensional data, but bucketing into 25 buckets immediately reduces the complexity of the observation space to 25-dimensional counts. However, the authors have addressed most of my major concerns.

---

> ### Author Response · Authors · 2020-11-18
> **Response to Reviewer 5**
>
> We would like to thank R5 for recognizing the paper's strong points and the suggestions for improving it.
>
> 1. It is true that on occasion and in our attempt to fit everything in the space allowed, a lot of the details and explanations are compromised. This has hopefully been in large-part improved in the updated version of the main text as well as in the pre-existing and new Appendices. The real-part of the complex-valued model parameters $\lambda_j$ is responsible for  the rate of time decay whereas the imaginary-part for the harmonic fluctuations. We have added expanded explanations in Appendix B with regards to the properties of these processes as well as a justification for the parameter choices.
> To demonstrate the utility of the extension in the  complex domain, we have added in Appendix H.1 the results obtained by  a method that makes use only of  real-valued variables.
>
>
> 2. We would like to clarify the fine-scale model  and the associated observables. The system consists of single-species, interacting particles. The label of each particle is not important as they are indistinguishable. Predictions of interest pertain to first-, second- and higher-order interaction statistics such as the ones reported in the main text and in Appendix F. The likelihood model should therefore possess permutation-invariance. This is achieved with the choice of physically-motivated variables $\boldsymbol{X}$ i.e. the discretized particle density and the multinomial distribution.
> We have updated the text to indicate this more clearly and moved the details  of the numerical experiments to the main text.
>
> 3. The (prior) dynamics of the latent processes $z_{t,j}$ are explained in more more detail in the expanded Appendix B and the choice of parameters is justified. We have added numerical illustrations in the new  Appendix H where we demonstrate that fitting a flexible dynamical model (even when retaining the physically-motivated variables $\boldsymbol{X}$) can lead to good short-term predictions but in the longer time-horizon can fail dramatically leading to "unphysical" predictions. This could also be problematic when the trained model is used to make predictions with new initial conditions where the chaotic nature of the nonlinear dynamics can lead to significant errors even for shorter time horizons.
>
>
> - We have also included comparisons with variations of the method where the latent variables $\boldsymbol{z}_t$ lie  on the real-axis instead of the complex-plane as well as with deterministic and probabilistic  Koopman-based methods. We note that the latter generally require the definition of an encoder in order to identify some feature functions (also called, observables) and an associated finite-dimensional approximation of the Koopman operator. This gives rise to slew of model selection issues which we circumvent in our implementation with a fully generative model in which case the aforementioned encoder  becomes that posterior that is learned consistently with the decoder.  In the same settings illustrated in the main text and for the same amount of training data, we observe that the aforementioned alternatives produce predictive estimates of reduced accuracy and cannot capture the equilibrium density of the system. We hope that the results provided constitute sufficient evidence of the requested ablations.
>
>
>
> - Clarification: Predictions are obtained by using MAP estimates of the parameters $\theta$ but the (approximate) posteriors of state-variables $\boldsymbol{z}_t, \boldsymbol{X}_t$. We refer to this formulation as a hybrid  Bayesian approach as we do not infer the whole posterior of the model parameters $\boldsymbol{\theta}$.
>
> - Additional feedback: We have attempted to clarify such definitions in order to make the paper accessible to as wide of an audience as possible. We  thank the reviewer  for pointing out the typos.

---

### Author Response · Authors · 2020-11-18
**Revised Paper Uploaded**

We would like to thank the reviewers for their valuable feedback and have incorporated their suggestions in an updated version of our paper.
A detailed response to the reviewers' comments can be found below their reviews.

---

### Decision · Program_Chairs · 2021-01-07
**Final Decision**

**Decision:**

Accept (Poster)

**Comment:**

The consensus of the reviews is to accept the paper. I agree.

Reviewers highlighted many strengths, including a compelling main idea:
* R5: "The paper presents an interesting and motivating case for Bayesian inference in probabilistic generative models: a problem that has inherent uncertainty along with the ability to incorporate domain knowledge that can reduce the inference complexity."
* R3: "Overall, the idea is interesting and supported by correct mathematical derivations and experimental proofs of concept."
* R4: "the generative approach is novel. Adding domain knowledge is relevant and significant when dealing with real world applications"

As well as compelling experiments, substantially improved in the discussion period:
* R1: "The authors have shown some promising results in modeling particle dynamics."
* R5: "The addition of Appendix H, in my opinion, considerably strengthens the paper's story and case for acceptance. [... T]he authors have addressed most of my major concerns."

And clear writing:
* R5: "In general, the paper is well written (apart from some higher-level structural issues discussed below) and the notation is clear and unambiguous."
* R4: "The paper is very well written, clear"

The main weaknesses highlighted were in experiments (lacking good baselines, as well as ablations), and in discussing some choices in the model's construction. These were effectively addressed in the discussion (though R5 still points to some places that could be improved).